# Randomised controlled pilot trial of an exercise plus behaviour change intervention in people with multiple sclerosis: the Step it Up study

Sara Hayes,[1] Marcin Kacper Uszynski,[1,2] Robert W Motl,[3] Stephen Gallagher,[4] Aidan Larkin,[2] John Newell,[5] Carl Scarrott,[5,6] Susan Coote[1]

## ABSTRACT

**Objective** To investigate feasibility of multiple sclerosis (MS) exercise guidelines for inactive people with MS (PwMS) and to examine preliminary efficacy for walking. To investigate effect of augmenting that intervention with education based on social cognitive theory (SCT).

**Design** Pilot multicentre, double-blind, randomised, parallel, controlled trial.

**Setting** Community-delivered programme.

**Participants** Sixty-five physically inactive PwMS walked independently, scored 0–3 on the Patient Determined Disease Steps Scale, had no MS relapse or change in MS medication in 12 weeks.

**Interventions** 10-week exercise plus SCT education (SCT) compared with exercise plus attention control education (CON).

**Outcome measures** Six-Minute Walk Test (6MWT), Timed Up and Go (TUG) test and Multiple Sclerosis Walking Scale-12 (MSWS-12).

**Results** 174 expressed interest, 92 were eligible and 65 enrolled (SCT, n=32; CON, n=33). The intervention was feasible and delivered as intended. 68% of SCT group and 50% of control group met the exercise guidelines after intervention. Using linear mixed effects models, intention-to-treat basis, there was insufficient evidence for difference between the groups over the trial (6MWT, p=0.30; TUG, p=0.4; MSWS-12, p=0.8). Using secondary analysis of a cohort with data for ≥3 assessments (SCT, n=21; CON, n=20), there was significant treatment effect favouring the intervention group (p=0.04) with mean effect for 6MWT 39.0 m (95% CI 2.26 to 75.73) at 12 weeks and 40.0 m (95% CI 2.3 to 77.8) at 36 weeks. Both groups improved significantly in 6MWT following 10-week intervention (SCT, mean Δ=83.02, SD=60.1, p≤0.01; CON, mean Δ=56.92, SD=73.5, p≤0.01), TUG (SCT, Δ=−0.70, SD=1.25, p≤0.01; CON, Δ=−0.54, SD=0.95, p≤0.01) and MSWS-12 (SCT, Δ=−8.03, SD=16.18, p=0.02; CON, Δ=−0.86, SD=18.74, p=0.81).

**Conclusions** A 10-week exercise programme based on the MS exercise guidelines for improving walking in previously inactive PwMS was feasible. There is marginal evidence of a treatment effect in favour of the exercise plus SCT intervention at 12 and 36 weeks.

**Trial registration number** NCT02301442; Results.

## Strengths and limitations of this study

► New evidence demonstrating the feasibility and preliminary efficacy of delivering a pragmatic, combined, community-based exercise and social cognitive theory education intervention for physically inactive people with multiple sclerosis (MS) based on the MS Exercise Guideline.

► The use of measures of fidelity, assessments of the target variables of the intervention (strength, fitness and physical activity) and both self-report and objective measures of walking mobility.

► Treatment fidelity was considered and evaluated, yet a limitation relates to the use of a 1-day training course for physiotherapists, in particular relating to the novel use of education techniques throughout the exercise programme.

► Attrition of participants between determining eligibility and starting the intervention; the long wait times meant that 29% of eligible participants were lost at this phase.

## INTRODUCTION

Walking limitations are the hallmark of multiple sclerosis (MS)[1] and people with MS (PwMS) report that walking limitations are a significant concern.[2] Indeed, walking limitations have been associated with change in occupation due to MS and occupational disability[3] and influence a range of other outcomes such as cognition and depression.[4] Exercise training remains the cornerstone therapeutic intervention for the management of walking limitations in MS. Many studies report positive effects from a range of exercise interventions as summarised in recent reviews[5] and meta-analyses[6 7] that confirm combined aerobic and resistance exercises can improve both walking speed and walking endurance.

The recent exercise guidelines recommend aerobic exercise twice a week and resistance exercise twice a week as the minimum target

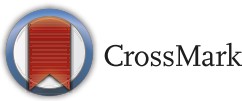

For numbered affiliations see end of article.

**Correspondence to**
Dr Susan Coote;
Susan.Coote@ul.ie

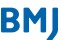

for improving walking outcomes among people with mild-to-moderate MS.[8] To that end, we demonstrated using a pragmatic, randomised controlled trial design, that 10 weeks of combined aerobic and resistance training delivered in groups in the community yielded positive improvements in Six-Minute Walk Test (6MWT).[9] Of concern, however, was that these improvements were not maintained at 3-month follow-up.[10]

The maintenance of long-term exercise behaviour change is not a problem that is unique to MS, and researchers have highlighted the need for inclusion of behavioural approaches based on theory for long-term behaviour change.[11] Social cognitive theory (SCT) has been most commonly investigated in MS and its domains of exercise self-efficacy and goal setting are consistently associated with physical activity (PA) behaviour.[12] We have reported improvements in PA and secondary outcomes including walking, from an SCT-based online intervention in MS,[13] and one study demonstrated that PA behaviour change was maintained 3 months after cessation of the program.[14] This education programme was originally designed based on a Randomised Controlled Trial (RCT) of a SCT-based exercise intervention delivered in older adults[15] and later modified and tested for MS.[16]

We designed a randomised controlled pilot trial called 'Step it Up'[17] that combined a group exercise programme with a theory-based education component for augmenting the effect of exercise on walking outcomes and sustaining these changes over time. The aim of this study was to investigate the feasibility of delivering the combined interventions by physiotherapists and to establish preliminary clinical efficacy for improving walking outcomes; secondary outcomes will be provided in a parallel publication. We delivered the same exercise programme to both groups and controlled for contact by comparing a structured SCT education programme with an attention control education programme and investigated whether adding the SCT education component would yield greater improvements in walking mobility and whether the improvements were maintained at follow-up. It was hypothesised that that the participants in the exercise and SCT-based intervention would achieve significantly more improvement in walking outcomes than the control group postintervention and that this improvement would be maintained at follow-up. The results of this trial will inform the design, particularly power analysis, of a definitive trial that provides Class 1 evidence (AAN).

## METHODS
### Study design
This was a multicentre, two-arm, parallel (1:1), double-blind, randomised controlled trial.

### Setting and participants
The participants were recruited through the MS Society of Ireland and via neurology clinics in three urban locations in Ireland. Details of the recruitment process are further detailed in the protocol paper.[17] Inclusion criteria were: (1) physician-confirmed formal diagnosis of MS, (2) aged 18 years or more, (3) Patient Determined Disease Steps (PDDS) Scale score of 0–3, (4) a sedentary lifestyle (<30 min of moderate to strenuous exercise 1 day or more per week over the last 6 months) and (5) willing to give written informed consent. Exclusion criteria included: (1) pregnancy, (2) MS relapse in the previous 12 weeks and (3) changes to MS medication or steroid treatment in the previous 12 weeks. Participants were sent the consent form in advance of the baseline assessment, and written consent was obtained in person.

### Randomisation and blinding
Participants were randomly allocated into the exercise plus SCT-based intervention or the exercise plus contact control education intervention. Random allocation procedures have been previously outlined[17] and were adhered to. JN generated the random allocation sequence, the SH enrolled participants and SC assigned participants to interventions. The outcome assessor (SH) was blind to allocation throughout the study as was the statistician CS during the analysis. All participants were informed that the study aimed to examine the effect of combining exercise and education, and therefore were blinded regarding group allocation.

### Screening questionnaire
Potential participants were screened for eligibility for this study using a questionnaire that included the PDDS Scale,[18] confirmation of formal MS diagnosis and questions regarding PA levels. The PDDS scale contains a single item for measuring self-reported neurological impairment on an ordinal level from zero (Normal) to eight (Bedridden). Scores from the PDDS are linearly and strongly related with physician-administered Expanded Disability Status Scale (EDSS) scores.[18]

### Outcome measures
Outcome measures were conducted preintervention (week 1), postintervention (week 12) and at 24-week and 36-week follow-up.

### Demographic and clinical information
Participants provided details regarding age, gender, level of formal education, time since diagnosis of MS, duration of symptoms of MS, falls history, exercise history, marital status and employment status. Additionally, a researcher formally trained in the use of the EDSS (SH) administered the EDSS to all participants at baseline. The EDSS quantifies MS disease progression and is commonly the standard that other outcome measures are compared against.[19] It consists of functional systems subscales and a total score which is an ordinal rating ranging from 0 (normal neurological status) to 10 (death due to MS). MS diagnosis according to the McDonald or Poser criteria was confirmed from the participant's consultant neurologist.

## Primary outcomes

The primary outcome was walking mobility at week 36. This was measured using the 6MWT as the primary endpoint. The participants were instructed to walk as quickly and as safely as possible for 6 min on a 10-metre track. The 6MWT has demonstrated excellent test-retest reliability and concurrent validity among people with mild-to-moderate MS.[20]

We further used the Timed Up and Go test (TUG) and the Multiple Sclerosis Walking Scale-12 (MSWS-12). The TUG has demonstrated excellent test-retest reliability for people with mild MS[21] and the MSWS-12 has demonstrated excellent internal consistency,[22 23] test-retest reliability[24] and concurrent validity in PwMS.[25]

## Adherence

Adherence to the intervention was documented throughout the 10-week intervention via exercise logs (see online supplementary appendix 1 Exercise Logs). The exercise logs captured attendance at the exercise classes and home exercise sessions. Over the 10-week intervention, 44 total sessions were made available to the participants. This included six exercise classes with strengthening and coaching/education components, four coaching phone calls, 14 prescribed home strengthening sessions and 20 prescribed home walking sessions.

We further evaluated adherence to the exercise component by evaluating the effect on strength, fitness and PA. The five times sit to stand test (5xSTS)[26] (time to complete five sit to stand repetitions in seconds) measured lower extremity muscle strength. The Modified Canadian Aerobic Fitness Test[27] measured fitness and was calculated using following equation; 10 x (17.2 + (1.29 x $O_2$ cost of last stage) – (0.09 x body mass in kg) – (0.18 x Age)). The Health Index of the Godin Leisure-Time Exercise Questionnaire[28] measured PA behaviour. These measures and associated psychometric properties have been described in the trial protocol.[17]

## Interventions

The content of the interventions delivered in both arms of this RCT has been outlined in detail in the protocol paper[17] and is further outlined in online supplementary appendix 2 (TIDieR checklist). The exercise intervention was common to both groups and was delivered by physiotherapists. The aim of the exercise component was to progressively increase the intensity of both aerobic and strengthening activities to enable the participants to reach the published exercise guidelines for people with mild-to-moderate MS,[8] and has been previously described.[17] Over the 10-week programme, participants attended the group exercise class (see online supplementary appendix 3 Exercises) at community venues on six occasions, supplemented with a telephone coaching call in the weeks without classes (intervention weeks 4, 6, 7 and 9). After each of the group exercise classes, the control group received an education session about topics unrelated to PA behaviour, for example, diet, vitamin D,

sleep, temperature and hydration and immunisations and vaccinations. The exercise plus SCT-based intervention group received the same exercise intervention as the control group (as described in the previous section). This group also received a similar duration of education based on the principles of SCT for health behaviour change, namely: self-efficacy, outcome expectations, goal-setting, barriers and benefits and has been previously described.[17] The SCT intervention was designed to enable continued exercise behaviour and after the 10-week intervention the participants in both groups received structured phone calls from the intervention physiotherapists at weeks 16, 20 and 36. These telephone calls consisted of direct questions about the frequency, intensity, type and duration of exercise participants had completed and whether they had experienced any adverse events or relapses. Additionally the SCT group was coached using the principles of that educational component.

## Treatment fidelity

All of the physiotherapists who delivered the intervention or control group sessions were provided with a 1-day training course on the delivery of the intervention for their group, directly related to the manual of operating procedures.[17] The intervention was delivered at three sites over the course of the study by eight physiotherapists broadly representative of those working in primary care. Continued support from the research centre was available if additional training was needed. The fidelity of the physiotherapists' sessions, including both exercise and SCT components, was monitored by randomly allocated video and audio recording of at least one of the intervention sessions. An independent assessor compared the content of the intervention manuals with the video or audio recordings.

## Statistical analysis
### Sample size

Consistent with data from a large international study,[29] it was hypothesised that the effect of the intervention would yield an average improvement in 6MWT distance of 36 m with an estimated SD of 48.2 m. In order to have 80% power (at the 5% significance level) to detect such a difference in mean improvement in 6MWT over the study period between groups, a sample of size 62 randomised equally to two arms (ie, 31 per arm) was used to inform the target sample size for this pilot study. The intention was to recruit 72 participants to account for dropout and to run the group interventions once sufficient people in that region were eligible. Recruitment in regions was not uniform and participants became ineligible while waiting for others to be recruited. Recruitment was better than intended and continued to 92 eligible participants resulting in 65 participants starting the intervention.

Suitable numerical statistics and graphical summaries were used to describe characteristics of the sample at baseline and to assess the validity of any distributional assumptions needed for the formal analysis. All

tests of significance were two-sided and conducted at an alpha=0.05 level of statistical significance. An exploratory paired t-test between baseline and each of the week 12, 24 and 36 follow-ups was conducted, providing a summary of the effects of the estimated treatment and control from the raw data. These 'unadjusted' results do not account for the patient covariates and repeated measurements. We also quantified and compared the magnitude of change in walking measures using Hedges' G effect sizes and associated 95% CIs using Cohen's conventions for effect sizes (0.2 small, 0.5 moderate, 0.8 large). For each outcome measure, the mean baseline to postintervention and 3-month and 6-month change for the control condition was subtracted from the mean baseline to postintervention and 3 and 6 month change for the intervention condition and divided by the pooled baseline SD.[30] Effect sizes were calculated such that greater improvements in outcomes in the intervention group compared with the control group resulted in positive effect sizes.

The statistical modelling compared differences in the response variables (6MWT, TUG and MSWS scores) between the two intervention arms at each of the three postintervention follow-ups while correcting for the baseline measurements for each participant. A linear mixed model for a continuous response over time due to the two interventions, while adjusting for participant-specific covariates and factors, namely 6MWT at baseline, age, gender, time since diagnosis and MS type (ie, benign, primary progressive and relapsing-remitting). Treatment and time (and their interaction) were specified as fixed effects, centre (three levels) and subject (nested in centre) as random effects in order to account for homogeneity within centre and within subject correlation over time. Initially a model containing the main effects of the treatment, time and a treatment-by-time interaction was specified in order to test whether there is evidence that the treatment effects varies over time. If the interaction was deemed unnecessary (using a likelihood ratio test), the model was refitted excluding the interaction term, so the treatment effect was then constant over time. Two separate analyses were carried out. First, following an intention-to-treat principle in which all 65 patients who remained eligible to participate were considered. In the secondary analysis, a smaller cohort of 52 patients are analysed, who were identified to have closely adhered to the programme by having attended at least two of the three follow-ups. All models were fitted in R 3.2.0 using the lme4 and lmerTest packages. Model diagnostics involved suitable plots of the residuals.

## RESULTS
### Participant sample
One hundred and seventy-four PwMS contacted the trial centre and were screened for inclusion over the phone between September 2013 and May 2014. Eighty-two people were excluded as per the selection criteria (figure 1) and recruitment ceased when 92 people were randomised to either of the trial arms. Between time of randomisation and initiation of the intervention, 27 eligible participants either became ineligible or were unable to participate. One participant was not treated as randomised (two acquaintances had been randomised to the other group and they wanted to exercise with them). Sixty-five participants commenced the intervention (SCT group, n=33; CON group, n=32). In the SCT group, four participants discontinued the intervention and 12 were lost to follow-up at 36 weeks. In the CON group, three participants discontinued the intervention and 10 were lost-to-follow up at 36 weeks. Following the 10-week intervention, overall attrition was 17% and at the 36-week follow-up assessment attrition was 34%. Reasons for discontinuing the intervention and loss to follow-up are outlined in figure 1. Baseline characteristics for both groups are shown in table 1.

### Treatment fidelity
An independent person to the intervention (PO'S) used the manual of operating procedures to check if the required content of the programme (both exercise and SCT/attention control education components) was delivered as intended. In both trial arms, 100% of the content of the supervised sessions were implemented as described in the intervention manual.

### Feasibility—Exercise Logs
The development of hip pain by one participant in the CON group was the only related adverse event reported by participants in both trial arms during the completion of the 10 week intervention. The SCT and the CON groups completed an average of 33.2 of 44 available sessions (75.5%) and 32.0 sessions (72.6%), respectively. The proportion of sessions completed is presented in figure 2, where the lowest number of sessions was in week 7 when participants were exercising independently without a class for a second consecutive week. Among the 53 participants who provided detailed exercise logs, 17 (68%) of the SCT group and 14 (50%) of the CON group were exercising at the minimum recommended by the exercise guidelines by the end of the 10-week intervention. The reasons for not meeting the guideline included: walking less than 30 min twice per week (SCT, n=3; CON, n=1), walking only once per week (SCT, n=2; CON, n=5) and doing only one set of each resistance exercise per week (SCT, n=2; CON, n=4).

In order to further evaluate the adherence to the intervention we investigated the change in strength, fitness and PA in order to evaluate whether the intervention changed these intended parameters. Table 2 presents the raw data and unadjusted comparisons. For both groups, there were significant improvements in PA and strength from weeks 1 to 12. There was a tendency for aerobic fitness scores to increase, but this change was not statistically significant.

### Walking mobility
The mean (SD) scores for the 6MWT, TUG and MSWS-12 at weeks 1, 12, 24 and 36 for participants in the exercise

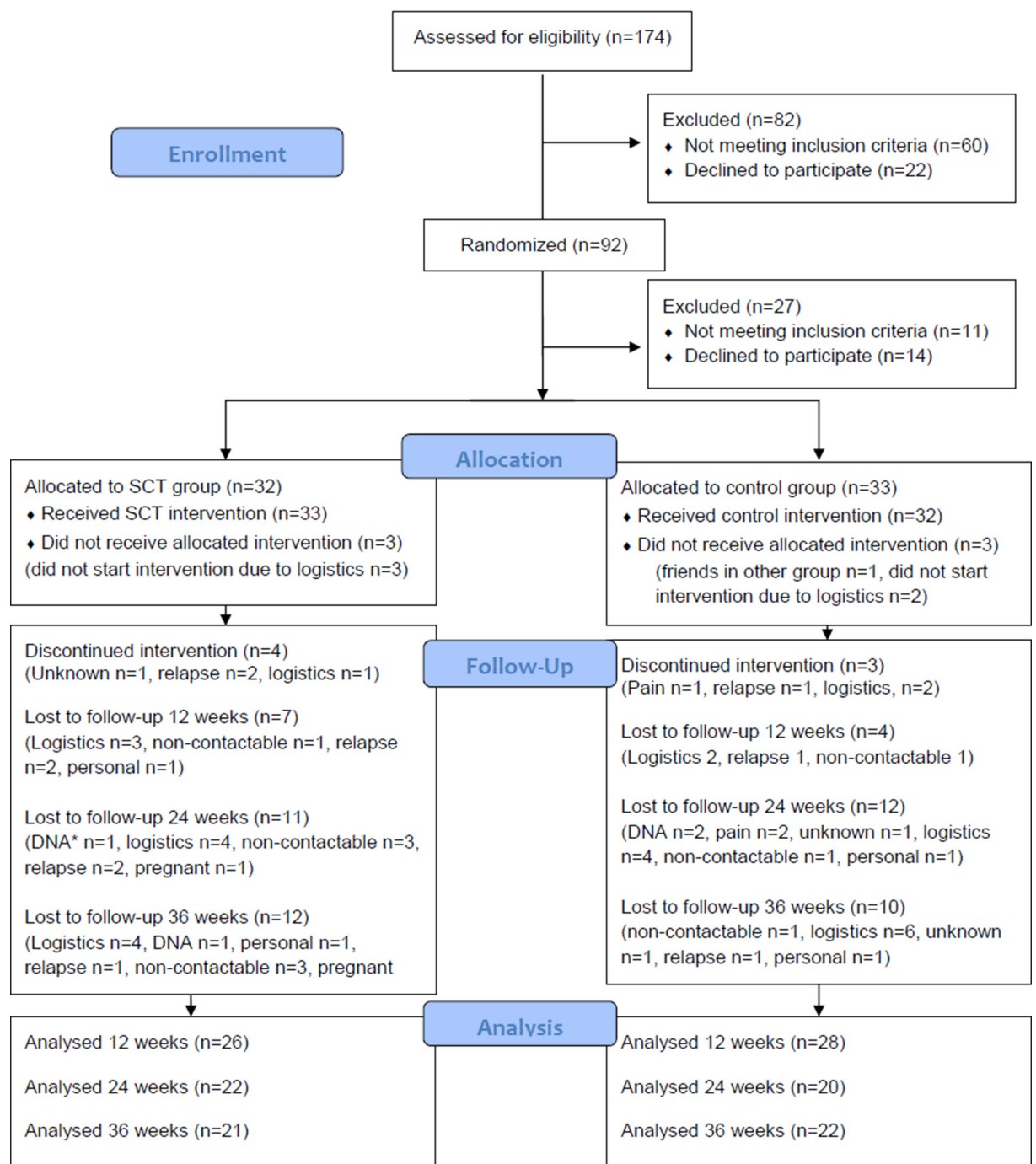

**Figure 1** CONSORT flow diagram.

plus SCT and exercise plus education control groups are presented in table 3. Figure 3 shows the results of the estimated treatment effects on 6MWT, TUG and MSWS-12, as per intention-to-treat and secondary analyses, respectively. The unadjusted, unstandardised mean changes from baseline, and 95% CIs and paired t-test results for both groups are presented in table 4 along with Hedges' G effect sizes. Both groups demonstrated an improvement in the primary outcome, 6MWT and secondary outcome MSWS from weeks 1 to 12 and at 24-week and 36-week follow-up. For TUG the result are a little more mixed, with evidence of an improvement in both groups from

weeks 1 to 12 which diminishes in the control group by week 36 but a persistent significant difference is observed in the education with SCT group from baseline to weeks 24 and 36.

The linear mixed models results in table 5 shows that using an intention-to-treat analysis there was no evidence of a significant treatment effect in favour of the exercise plus SCT compared with the exercise only group for regarding 6MWT, TUG or MSWS scores. Figure 3 confirms the obvious significant effects of the exercise programme found above in the unadjusted paired t-test results, which is shown by the blue and red lines being

**Table 1** Clinical baseline characteristics in exercise plus SCT group (SCT) and exercise plus education control group (CON)

|  | SCT (n=33) | CON (n=32) |
|---|---|---|
| MS type | | |
| Benign | 3 | 1 |
| Primary progressive | 1 | 0 |
| Relapsing-remitting | 27 | 27 |
| Secondary progressive | 0 | 1 |
| Unknown | 2 | 2 |
| EDSS (median, IQR) | 3.3 (0.7) | 3.3 (0.7) |
| Years since diagnosis | 6.7 (5.7) | 7.0 (6.1) |
| Centre (n) | | |
| Cork | 10 | 9 |
| Galway | 8 | 10 |
| Limerick | 15 | 13 |
| Age | 43.3 (9.9) | 41.9 (9.3) |
| Gender (n) | | |
| Male | 4 | 6 |
| Female | 29 | 26 |

Data given as mean (SD) unless otherwise indicated.
EDDS, Expanded Disability Disease Scale; MS, multiple sclerosis.

well above the black 'no effect' line when the sample uncertainty conveyed by the corresponding CIs are taken into account. But figure 3 also confirms lack of evidence for an additional effect of the SCT over the usual exercise programme, which is shown by the widely overlapping CIs between the treatment and control groups.

A secondary analysis was completed with participants who attended at least two of the three follow-up assessments (SCT n=25, CON n=27). Table 3 presents the mean (SD) scores for the 6MWT, TUG and MSWS-12 at weeks 1, 12, 24 and 36 for participants in the SCT and control groups using secondary analysis. For 6MWT, the SCT group had a marginally more positive outcome, with statistically significant treatment effects evident at weeks

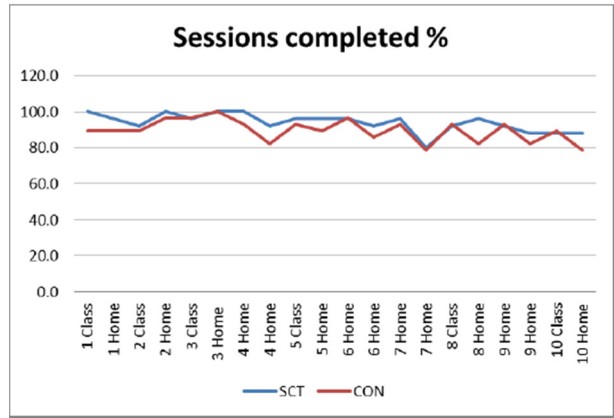

**Figure 2** Proportion of participants completing sessions (Exercise Diary data).

12 and 36 (table 5). Using this secondary analysis, there was no evidence of a treatment effect in favour of the SCT group as compared with the CON group regarding the TUG or MSWS-12 scores.

## DISCUSSION

This pilot RCT investigated the feasibility and preliminary efficacy of the Step it Up programme, a 10-week aerobic and strengthening programme that aimed to enable physically inactive PwMS to exercise according to the recent MS exercise guidelines.[8] We investigated whether embedding an evidence-based exercise programme within a structured SCT-based education programme resulted in improved and more sustained walking outcomes compared with an exercise plus attention control education intervention. To our knowledge, this is the first study to examine the effect of enabling inactive people to meet the minimum recommended dose of the MS exercise guidelines and examine the effects on walking mobility as a primary end-point.

The intervention protocol was feasible and results demonstrated significant improvements in walking mobility following the intervention in both groups. There was a moderate effect (Hedges' G 0.50) at 36-week follow-up in favour of the SCT group for 6MWT. The effect for the SCT group was also greater at 12-week and 36-week follow-up for the primary outcome, 6MWT, using the secondary analysis which included only patients who adhered to the programme (as defined by having attended at least two of the three follow-ups). Recruitment was successful and over 9 months at three centres, we recruited more than our target of 62 participants (92 eligible participants). The largest point of attrition was while participants waited for enough people to run the group in that region. In the future, recruiting from the largest city in Ireland for a definitive RCT will enable greater numbers to be recruited more quickly and should minimise this attrition at this point in the trial. Retention across the intervention period was good and the attrition rate (17%) was similar to other exercise interventions in people with depression[31] and slightly higher than the average of 15% in a review of exercise trials in MS.[32] While the level of participant attrition in the current programme is greatly improved from our previous community based exercise RCT,[9 10] measures such as recruiting a dedicated study coordinator to provide more frequent interactions with participants in the definitive trial will be explored to further enhance retention at follow-up. The addition of booster intervention sessions after the completion of the 10-week intervention will also be explored in the future definitive trial.

The intervention was delivered by physiotherapists who attended a 1-day training session and treatment fidelity findings suggest that this approach was successful as the interventions were delivered as intended; further training and support may increase the success of the intervention in future. Participants completed on average 73%–75%

**Table 2** Raw data and unadjusted comparisons of change in secondary outcomes from week 1 to week 12 in exercise plus SCT group (SCT) and exercise plus education control group (CON)

| | | Week 1 Mean (SD) | Week 12 Mean (SD) | Mean change from week 1 to week 12 (95% CI) p Value |
|---|---|---|---|---|
| Godin Health Index | SCT | 3.03 (6.19) | 12.48 (11.15) | 9.85 (5.46 to 14.23) p<0.01 |
| | CON | 1.88 (4.88) | 16.07 (21.12) | 12.92 (4.96 to 20.89) p<0.01 |
| Five Times Sit to Stand | SCT | 11.48 (2.7) | 9.78 (2.18) | −1.51 (−2.42,−0.60) p<0.01 |
| | CON | 10.8 (2.6) | 9.43 (1.93) | −1.55 (−2.30 to −0.79) p<0.01 |
| Aerobic Fitness Score | SCT | 295.72 (54.61) | 309.12 (53.78) | 8.58 (−6.86 to 23.98) p=0.26 |
| | CON | 313.56 (59.02) | 331.29 (51.57) | 10.54 (−6.29 to 27.37) p=0.21 |

of possible sessions suggesting that the protocol is feasible for participants with minimal impairment due to MS. We collected data from exercise logs for demonstrating adherence with the exercise programme. The exercise logs were returned by 82% of participants and used to ascertain whether participants were meeting the MS exercise guideline at the end of the intervention period. It is interesting to note that a greater proportion of participants in the SCT group (68% vs 50%) progressed to meeting the guidelines. Measures to further enhance completion and return of logs (such as offering them in alternative electronic formats) in the future definitive trial are needed.

We further confirmed adherence to this aerobic and strengthening intervention by investigating its effects on strength, fitness and PA. Both 5xSTS and Godin Health index increased significantly and the Aerobic Fitness Scale (AFS) showed a tendency to improve providing evidence that the exercise intervention met its intended outcomes. Collectively, we believe that the exercise log data combined with fitness and PA outcomes support the successful manipulation of exercise behaviour with in both

trial arms. Based on the data on recruitment, retention, feasibility and preliminary efficacy of this group exercise and SCT education intervention, we propose to progress to a definitive intervention. To do this, a sample of 49 (for a difference between groups of 39 m, assuming a SD of the change score at 36 weeks of 67.85, 80% power, 0.05 significance level) in each group would be needed and we therefore plan to recruit across these three centres again and to add a fourth centre in the largest city in Ireland to minimise attrition.

Importantly, both groups improved significantly in the primary outcome, 6MWT, following the intervention. This improvement in 6MWT is consistent with a recent systematic review of exercise studies that found a significant improvement in walking endurance.[7] We note that the mean improvement in the SCT group of 80 m and of 60 m in the control group far exceeded the value for the clinically important change of 26.1 m proposed by Baert et al.[29] Both groups improved more than that reported by Carter et al[33] in their exercise plus education group, and the magnitude of improvement is more consistent with the improvements noted in a recent community-based

**Table 3** Mean (SD) walking mobility outcomes at weeks 1, 12, 24 and 36 in exercise plus SCT group (SCT) and exercise plus education control group (CON)

| Outcome variable | Week 1 | | Week 12 | | Week 24 | | Week 36 | |
|---|---|---|---|---|---|---|---|---|
| | SCT | CON | SCT | CON | SCT | CON | SCT | CON |
| | Intention-to-treat analysis | | | | | | | |
| 6MWT | 445.2 (68.8) | 482.0 (72.0) | 527.4 (91.1) | 547.1 (96.0) | 492.8 (73.5) | 504.9 (76.9) | 515.8 (91.0) | 528.0 (93.2) |
| TUG | 7.06 (1.61) | 6.51 (1.36) | 6.27 (1.45) | 5.81 (1.08) | 6.23 (1.26) | 6.00 (0.98) | 5.93 (1.33) | 5.96 (1.20) |
| MSWS-12 | 38.0 (28.0) | 33.3 (24.8) | 29.6 (22.2) | 30.8 (21.3) | 31.9 (22.1) | 26.3 (21.5) | 32.6 (23.4) | 27.9 (21.9) |
| | Secondary analysis | | | | | | | |
| 6MWT | 434.6 (65.2) | 474.4 (69.6) | 524.2 (96.7) | 535.2 (88.0) | 496.2 (73.7) | 504.9 (76.9) | 515.8 (91.0) | 528.0 (93.2) |
| TUG | 7.08 (1.73) | 6.65 (1.36) | 6.43 (1.46) | 5.87 (1.13) | 6.30 (1.25) | 6.00 (0.98) | 5.93 (1.33) | 5.96 (1.20) |
| MSWS-12 | 38.2 (26.7) | 31.9 (22.6) | 29.7 (22.6) | 32.6 (21.0) | 31.9 (22.1) | 26.3 (21.5) | 32.6 (23.4) | 27.0 (21.8) |

MSWS-12, Multiple Sclerosis Walking Scale-12; 6MWT, Six-Minute Walk Test; TUG, Timed Up and Go.

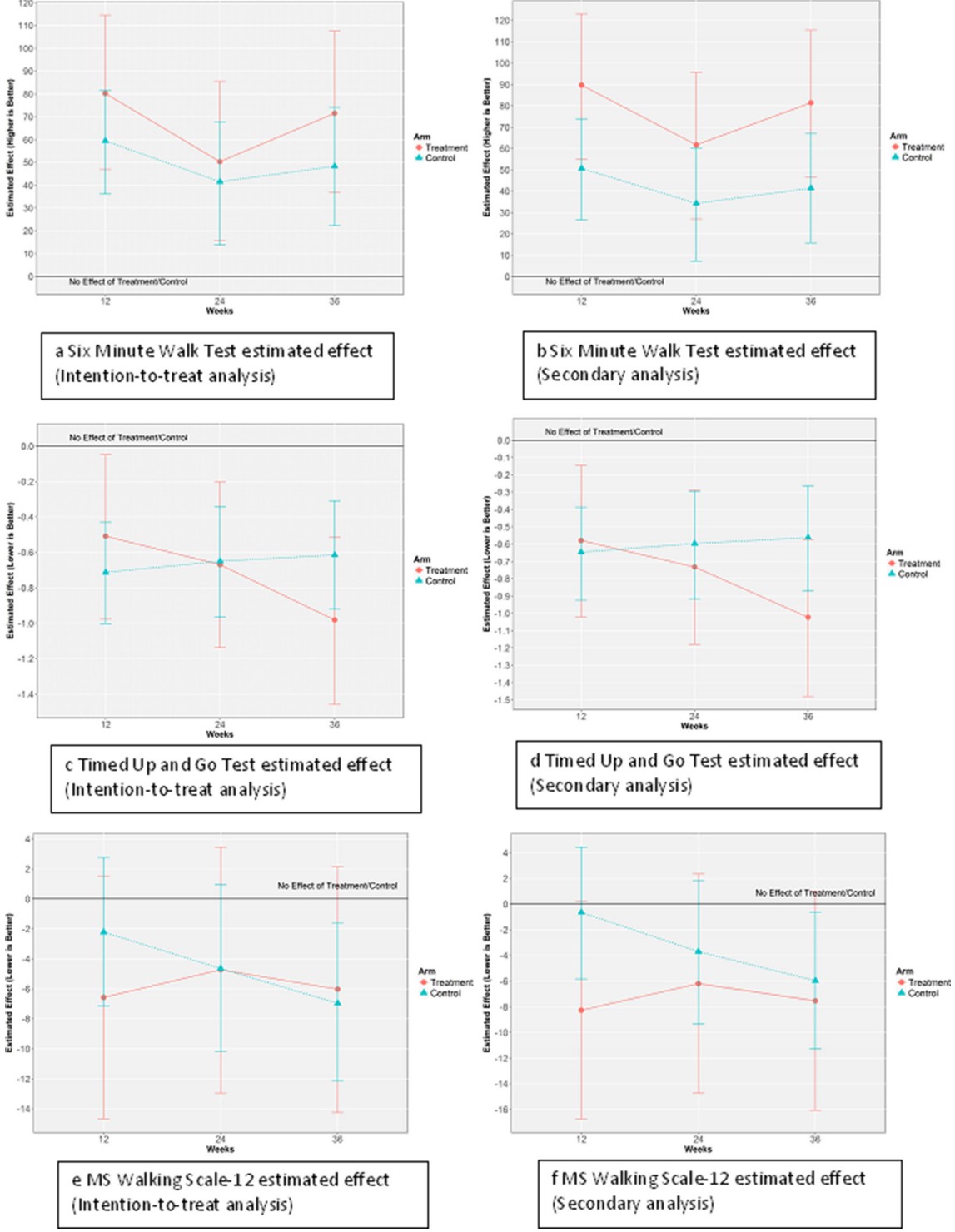

**Figure 3** Estimated effects using intention-to-treat and secondary analyses. MS, multiple sclerosis.

intervention among people with moderate-severity MS.[34] We further note that the current physically inactive sample of PwMS with an average age of 42 had 6MWT of 445 m at baseline that was less than that of a reference sample aged 70–80 years who walked an average of 514 m[35]. This confirms the significant walking impairments for inactive people with mild disability with MS and importantly demonstrates positive improvements due to the Step it Up exercise intervention. Interestingly, the SCT group

but not the CON group improved in their self-reported walking impairment (MSWS-12) and the magnitude of the change in 6MWT distance may have influenced that finding. Both groups however improved in walking speed and maintained that improvement at 36-week follow-up with a small-moderate effect size in favour of the SCT group demonstrated for TUG.

Of note, through the secondary analysis including participants who participated in at least two follow-up

**Table 4** Unadjusted comparisons of change in walking measures from week 1 to weeks 12, 24 and 36 in exercise plus SCT group (SCT) and exercise plus education control group (CON)

| | Mean change week 1 to week 12 (95% CI) p Value | | | Mean change week 1 to week 24 (95% CI) p Value | | | Mean change week 1 to week 36 (95% CI) p Value | | |
|---|---|---|---|---|---|---|---|---|---|
| | SCT | CON | Hedges' G (95% CI) | SCT | CON | Hedges' G (95% CI) | SCT | CON | Hedges' G (95% CI) |
| 6MWT | 83.02 (58.74 to 107.29) p<0.01 | 56.92 (28.43 to 85.41) p<0.01 | 0.37 (−0.12 to 0.86) | 55.97 (32.12 to 79.84) p<0.01 | 34.2 (13.43 to 54.97) p<0.01 | 0.31 (−0.18 to 0.80) | 82.18 (50.90 to 113.45) p<0.01 | 46.87 (18.57 to 75.17) p<0.01 | **0.50** (0.01 to 0.96) |
| TUG | −0.70 (−1.20 to −0.19) p<0.01 | −0.54 (−0.91 to −0.17) p<0.01 | 0.11 (−0.59 to 0.38) | −0.79 (−1.19 to −0.38) p<0.01 | −0.74 (−1.13 to −0.35) p<0.01 | 0.03 (−0.52 to 0.45) | −1.23 (−1.68 to −0.78) p<0.01 | −0.57 (−0.98 to −0.16) p<0.01 | 0.44 (−0.05 to 0.93) |
| MSWS-12 | −8.03 (−14.43 to −1.63) p=0.02 | −0.86 (−7.99 to 6.27) p=0.81 | 0.27 (−0.22 to 0.76) | −6.43 (−12.10 to −0.77) p=0.03 | −2.88 (−11.41 to 5.64) p=0.49 | 0.13 (−0.35 to 0.62) | −8.62 (−15.90 to −1.34) p=0.02 | −5.60 (−13.84 to 2.64) p=0.17 | 0.11 (−0.37 to 0.60) |

Bold text indicates moderate effect size.
MSWS-12, Multiple Sclerosis Walking Scale-12; 6MWT, Six-Minute Walk Test; TUG, Timed Up and Go.

**Table 5** Estimated treatment effects at weeks 12, 24 and 36 in primary outcome

| | Estimate of difference between SCT and Control | SE | 95% CI | p Value |
|---|---|---|---|---|
| **Intention-to-treat analysis** | | | | |
| 6MWT | | | | |
| Week 12 | 22.70 | 19.00 | (−15.14 to 60.50) | 0.23 |
| Week 24 | 11.80 | 20.40 | (−28.77 to 52.36) | 0.56 |
| Week 36 | 27.42 | 20.35 | (−13.06 to 67.90) | 0.18 |
| TUG | | | | |
| Week 12 | 0.069 | 0.236 | (−0.402 to 0.541) | 0.77 |
| Week 24 | −0.132 | 0.250 | (−0.630 to 0.365) | 0.60 |
| Week 36 | −0.457 | 0.252 | (−0.960 to 0.045) | 0.08 |
| MSWS-12 | | | | |
| Week 12 | −4.91 | 4.47 | (−13.82 to 4.00) | 0.28 |
| Week 24 | −0.59 | 4.69 | (−9.91 to 8.73) | 0.90 |
| Week 36 | 0.38 | 4.57 | (−8.71 to 9.47) | 0.93 |
| **Secondary analysis** | | | | |
| 6MWT | | | | |
| Week 12 | 39.00 | 18.44 | (2.26 to 75.73) | 0.04 |
| Week 24 | 27.44 | 19.23 | (−10.82 to 65.70) | 0.16 |
| Week 36 | 40.03 | 18.97 | (2.27 to 77.79) | 0.04 |
| TUG | | | | |
| Week 12 | 0.204 | 0.255 | (−0.306 to 0.713) | 0.43 |
| Week 24 | −0.020 | 0.261 | (−0.542 to 0.502) | 0.94 |
| Week 36 | −0.367 | 0.262 | (−0.890 to 0.156) | 0.17 |
| MSWS-12 | | | | |
| Week 12 | −7.63 | 4.65 | (−16.89 to 1.63) | *0.11* |
| Week 24 | −2.50 | 4.78 | (−12.01 to 7.02) | *0.60* |
| Week 36 | −1.57 | 4.69 | (−10.93 to 7.78) | *0.74* |

MSWS-12, Multiple Sclerosis Walking Scale-12; 6MWT, Six-Minute Walk Test; TUG, Timed Up and Go.

assessments, we demonstrated that adding a structured SCT education programme enhanced the effect on 6MWT distance following the 10-week intervention. This is important as it provides information on the preliminary effectiveness of the intervention and confirms the need to augment the retention strategies in the definitive trial. We propose greater training for the interventionists and greater use of telephone coaching in weeks without classes and between intervention and follow-up sessions. Importantly, both the improvement from baseline and the difference in between-group effects were maintained at 36-week follow-up providing new information on the ability to sustain effects after the intervention ceased. Interestingly, the effect was reduced at 24 weeks and participants reported that realising they had deteriorated at that assessment served as a prompt to resume their exercise after that assessment. The SCT education programme had six education sessions that targeted outcome expectancies, self-efficacy, goal setting and perceived barriers and benefits of exercise. The components are further consistent with a recent systematic review and meta-analysis of modifiable psychosocial constructs associated with PA in MS that confirmed self-efficacy, goal setting and outcome expectancies as significantly correlated with PA in MS.[36] One novel feature of the current trial is that the SCT education modules were delivered by physiotherapists with minimal training in delivery of behavioural interventions. These findings also support that delivering this SCT education intervention by physiotherapists in a group setting is both feasible and preliminary findings suggest that it may have superior outcomes to an attention control education intervention.

### Strengths and limitations

One of the strengths of this pilot RCT relates to the production of new knowledge around the sustainability of exercise interventions for PwMS. Building on the existing evidence base, we designed and delivered a SCT-based pragmatic physiotherapist-led community exercise. Results demonstrated the feasibility of the protocol among physically inactive people with mild MS and trends towards clinical efficacy for walking outcomes. The model of care outlined in this pilot study presents as a highly scalable intervention package for physiotherapists and other healthcare professionals working in primary care services or with third sector organisations (charities). Further study in the form of a definitive trial, including cost and clinical outcomes has the potential to have real policy implications for the provision of rehabilitation to PwMS. Further strengths relate to the use of measures of treatment fidelity, target variables of the intervention (strength, fitness and PA) and both self-report (MSWS) and objective (TUG and 6MWT) measures of walking. Additionally, in the context of an evidence base where PA interventions are often not theoretically based, a key strength of this RCT is the use of the SCT framework to design a behaviour-change intervention; building on the extensive work of the US partner in this trial.

One limitation is the attrition of participants between point of eligibility and allocation to the intervention. The large waiting times resulted in the loss of 29% of eligible participants at this point in the trial. Recruitment from larger urban areas with greater numbers of both MS clinics and PwMS is planned for the future definitive trial so that the numbers required to run group classes are met more quickly. A further positive is that we used pedometers and exercise logs to record the intensity and duration of the intervention; however, another limitation is that detailed exercise diaries were not returned for all participants. However, a return rate of 82% is acceptable and measures to improve this in the definitive trial will be considered.

### CONCLUSION

This pilot RCT aimed to investigate the feasibility and preliminary efficacy of enabling physically inactive PwMS to meet the MS exercise guidelines[8] through a group exercise and education, physiotherapist-led intervention. We further sought to investigate whether the theory-based SCT component was superior to an attention control education intervention. We found that recruitment was successful, though measures to improve retention in a future definitive trial are needed. Attrition over the intervention and follow-up periods were improved compared with our previous exercise trial.[9] The programme resulted in significant improvements in walking endurance and speed for both groups. There was a moderate effect (Hedges' G 0.50) for 6MWT at 36 weeks which is supported by a secondary analysis of those with data for three of four assessment points which demonstrated there was a significant effect in favour of the exercise plus SCT groups compared with the exercise plus control education group at weeks 12 and 36. This supports the preliminary sustained efficacy of the intervention and we propose progressing to a definitive intervention.

**Author affiliations**
[1]Department of Clinical Therapies, Faculty of Education and Health Sciences, Health Research Institute, University of Limerick, Limerick, Ireland
[2]Western Region, Multiple Sclerosis Society of Ireland, Dublin, Ireland
[3]Department of Physical Therapy, School of Health Professions, University of Alabama at Birmingham, Birmingham, Alabama, USA
[4]Department of Psychology, Faculty of Education and Health Sciences, Centre for Social Issues Research, University of Limerick, Limerick, Ireland
[5]HRB Clinical Research Facility and School of Mathematics, Statistics and Applied Mathematics, National University of Ireland, Galway, Ireland
[6]School of Mathematics and Statistics, University of Canterbury, Canterbury, New Zealand

**Acknowledgements** The authors would like to thank MS Ireland for their assistance with recruitment and running this trial. We would also like to thank Paraic O'Suilleabhain who assisted with data for this paper.

**Contributors** SH was a postdoctoral researcher on the trial, contributed to the design of the study, collected data, drafted the paper and approved the final version. MKU was a postdoctoral researcher on the trial, commented on drafts of the paper and approved the final version. RWM co-initiated the project and contributed to the design of the trial, drafted the paper and approved the final version. SG contributed to the design, delivery and evaluation of the trial, commented on drafts of the paper and approved the final version. AL contributed

to the recruitment strategy employed, commented on drafts of the paper and approved the final version. JN and CS were the statisticians on the trial, cleaned and analysed the data, commented on drafts of the paper and approved the final version. SC was the principal investigator for the study, co-initiated the project, contributed to the design of the trial, drafted the paper and approved the final version.

**Funding** This work is supported by the Irish Health Research Board Health Research Award, grant number: HRA_PHR/2013-264.

**Competing interests** None declared.

**Ethics approval** Ethics approval was given by the Faculty of Education and Health Science Research Ethics Committee, University of Limerick (2014_02_20_EHS), in addition to the Research Ethics Committees at the University College Hospital Galway, University Hospital Limerick and Cork University Hospital.

**Provenance and peer review** Not commissioned; externally peer reviewed.

**Data sharing statement** All data requests pertaining to the Step it Up trial should be made directly to susan.coote@ul.ie.

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
