## [Reviewer comments · BMJ Open]

ARTICLE DETAILS

TITLE (PROVISIONAL)	A randomised controlled pilot trial of an exercise plus behaviour change intervention in people with multiple sclerosis: the Step it Up study
AUTHORS	Hayes, Sara; Uszynski, Marcin; Motl, Robert; Gallagher, Stephen; Larkin, Aidan; Newell, John; Scarrott, Carl; Coote, Susan.

VERSION 1 – REVIEW

REVIEWER	Heesen University Medical Center, Hamburg, Germany
REVIEW RETURNED	02-Mar-2017

GENERAL COMMENTS	In this nice pilot RCT authors enrolled 65 participants on either a 10-week exercise training + SCT-based coaching or exercise with general MS lifestyle coaching. While both groups improved, there was no significant difference on the primary outcome, the 6MWT at week 12, 24 and 36. However, per protocol analysis (unclear number of patients) showed a significant effect at week 12 and 36. Authors conclude that a larger trial is justified based on these data. Authors address the highly relevant issue of health behaviour change and adherence to that change in relation to exercise. It is the first study explicitly addressing the added value of a coaching approach based on one of the major health belief models, the SCT. However, the study failed in meeting the primary endpoint. In this respect my major concern is if the sample size estimation was really justified. The cited paper is a multicenter cross-sectional study on walking measures. To me it is not clear how authors derived an estimate on the added effect on the 6MWT through the current intervention. In addition I wondered why only 50% of the control group and 68% in the SCT group did meet the exercise guideline recommendations when the 10 week intervention did end. Loss of FU was more than 30% from week 24 on which makes any longitudinal conclusion difficult. The discussion is too long. Minor points It is not clear what patients did after week 10. Did they train? Was this monitored? Page 5, l 10: I am not sure if walking limitation is really the primary reason for unemployment. Fatigue seems to compete with. Page 5, l 38: I am not sure if SCT can be regarded as „widely investigated“ in MS. In fact it has been a major focus of one of the coauthors.
---

	Page 5, l 44: Please indicate how long activity behavior change was maintained in the internet study Page 8, l 12: Was the 6MWT primary outcome at week 12, 24 and 26? Page 9, l 40: Was fidelity in relation to the SCT intervention or tot eh physiotherapy calöesses analysed? Page 13, ll 18 and table 2: Please clarify how aerobic fitness scores was derived and indicate how strength was assessed. Page 15 table 4: How do authors explain while in bothe groups intervention effects were less after 24 weeks compared to week 12 but increased again to week 36? It might be just due to the changing sample size. While a third oft the patients was lost at week 36 i would refrain from presenting data in parallel with that differing numbers. Otherwise authors might select only the cohort with complete data. Page 16 l 43: Please indicate number of patients for per protocol analysis Page 18, l 40: Topic oft the paper is not design of the next trial. Page 19, l 27: Where are data on walking speed given? Page 20, l 3: Based on which data do authors conclude that delivery oft the SCT education was adequate. Page 20, l 44: If authors do not report pedometer data it is difficult to judge these as a strength. Page 21, l 3: Conclusion omits the failure oft he primary endpoint.
--	--

REVIEWER	Johnny Collett Movement Science Group, OxINMAHR, Oxford Brookes University, UK
REVIEW RETURNED	09-Mar-2017

GENERAL COMMENTS	This is a useful study that seems well conducted and investigates an important area of MS rehabilitation. The study essentially tested the feasibility of adding a social cognitive theory behaviour change intervention to an exercise intervention of aerobic and strength training, using a randomised design, in sedentary people with MS that may have mobility problems. The study found it was feasible to deliver and evaluate the intervention. However, from the result, currently presented, it is difficult to evaluate the estimates of potential superior efficacy of incorporating SCT. My main concerns are: The analysis approaches used in the study are not clear (Below are example NICE definitions) The study does not seem to strictly follow intention-to-treat analysis 'An assessment of the people taking part in a trial, based on the group they were initially (and randomly) allocated to. This is regardless of whether or not they dropped out, fully adhered to the treatment or switched to an alternative treatment...' It appears the 27 people excluded after randomisation were not included and, according to figure 1 only 54 at week 12, 42 at week 24 and 43 at week 36 were included in analysis. Whilst, the 'intention-to-treat' approach used seems appropriate, clarification is needed in its description. More impotently I recommend removing the per- protocol analysis as feel it is not warranted in the context of this feasibility study. Per-protocol: 'A comparison of treatment groups in a trial that includes only those patients who completed the treatment they were originally allocated to. If done alone, this analysis leads to bias.'
---

The analysis reported is based on whether participants attended assessment or not, rather than completed the treatment as intended. I am not sure of the value of this analysis and would say it actual distracts from more valuable insights of the study.

The reporting of the outcome in results is a bit confusing and sometimes redundant. Most of the results appear to be focused on within group change, which does not greatly add to the literature. The interesting outcome results are the between group analysis and whilst difference between groups may be small, this would be expected given that both groups are exercising and outcomes are primarily walking. However, these small effects could potentially be important in terms of long term participation in exercise.

I feel the 6minute walk was not the optimal to evaluate the main aim of the intervention (added SCT) to enable people to meet MS exercise guidelines. In my opinion the most interesting and important outcome findings is that the intervention did indicate encouraging results for meeting guidelines at that at 10weeks (68% SCT v 50% control). Longer term follow up of this or other exercise physical activity measures) would have been interesting. Indeed the outcomes used in this study are mainly focused on impairment/performance. I believe discussion of this and the benefits of a longer term follow up in relation to future studies) is important (we pretty much know exercise will benefits walking, the challenge is longer term participation with the goal of sustained benefit to symptoms, health and QoL).

Table 2 should report between group differences (effect sizes and (95% CI)), not within.

I recommend mobility outcomes be reported in one table, basically table 3 with the addition of the between group treatment at effects of table 5 (reported as between group effect size with 95% CI)

In the trial registration primary outcome was all walking mobility measures whereas the study reports 6min walk as the primary. This should be made consistent.

Below are some further comments and suggestions

In the UK NIHR have distinct definition of feasibility and pilot studies (The title of the paper referring to pilot and objective being feasibility might be problematic for some people here)

Please provide numbers analysed in abstract as per CONSORT

In methods more detail could be added to study design section (page 6) (eg 2 arm, parallel (1:1))

Whilst, the sample size seems appropriate to fulfil the feasibility aims of the study, I am not quite sure how the authors derived their estimate and would like some clarification. It appears to be based on a between group mean difference in 6min walk of 36m +/- 48.2. I am not sure how this estimate was derived from the paper cited (eg the between group treatment effect), but also as a feasibility study I would expect a function of the study would be to provide estimates of effect rather than detect a difference. In addition a sample size of 72 people is stated in the protocol paper (to allow for drop outs). This should be made consistent, did the study not fully recruit (consideration for feasibility) or were dropouts less than expected (leading to protocol change).

JN generated the random allocation sequence yet JN was also blind to allocation, this is contradictory at face value and needs clarification?

Details of the track/area used for 6min walk (eg length of circuit, were people allowed to rest) would add

The authors report hip pain as the only adverse, was this this the only related AE, as the flow diagram indicates several relapse across groups (which could be considered AEs even though not

	related) Even with the Protocol paper I would find it very difficult to replicate the intervention, perhaps a description according to TIDieR could be added as a supplement. Making available the Manuel of operating procedures or the intervention Manuel would be very useful to anyone wishing to utilise the intervention. I feel the discussion would benefit from the addition of how this intervention may fit within current health or 3rd sector services, in particular in Ireland and perhaps wider.
--	--

VERSION 1 – AUTHOR RESPONSE

Reviewer: 1

Reviewer Name: Heesen

Institution and Country: University Medical Center, Hamburg, Germany Please state any competing interests or state 'None declared': Working as well in the exercise field. Some recent collaboration with authors of the study.

Please leave your comments for the authors below; In this nice pilot RCT authors enrolled 65 participants on either a 10-week exercise training + SCT-based coaching or exercise with general MS lifestyle coaching. While both groups improved, there was no significant difference on the primary outcome, the 6MWT at week 12, 24 and 36. However, per protocol analysis (unclear number of patients) showed a significant effect at week 12 and 36. Authors conclude that a larger trial is justified based on these data.

- Thank you for these positive comments, we have clarified the numbers in the per-protocol analysis -

Authors address the highly relevant issue of health behaviour change and adherence to that change in relation to exercise. It is the first study explicitly addressing the added value of a coaching approach based on one of the major health belief models, the SCT. However, the study failed in meeting the primary endpoint. In this respect my major concern is if the sample size estimation was really justified. The cited paper is a multicenter cross sectional study on walking measures. To me it is not clear how authors derived an estimate on the added effect on the 6MWT through the current intervention.

- We apologise for the lack of clarity in this element of the paper and have adjusted the text to provide a clearer justification. The purpose of this pilot trial was to obtain estimates to be used in a definitive study. We used the cited cross sectional study to generate initial sample size estimates for this pilot trial, as there was not a previous trial of this nature, and have reported this consistent with the justification required in the CONSORT extension for pilot trials

In addition I wondered why only 50% of the control group and 68% in the SCT group did meet the exercise guideline recommendations when the 10 week intervention did end.

- We have provided information as to what element of the guideline was lacking for those not meeting the guideline in the paper. We are conducting further moderation and mediation analysis of the effect on 6MWT which will be reported as a separate paper as this was not the main aim of this pilot trial.

Loss of FU was more than 30% from week 24 on which makes any longitudinal conclusion difficult.

- We agree, and hence will move forward to a fully powered definitive trial accounting for this level of attrition. We have modified our conclusion with the loss of persons over time.

The discussion is too long.

- We have edited the discussion, where appropriate, but were not provided with a specific source/section for reduction.

Minor points

It is not clear what patients did after week 10. Did they train? Was this monitored?

- This information has been added to the intervention section of the method.

Page 5, l 10: I am not sure if walking limitation is really the primary reason for unemployment. Fatigue seems to compete with.

- Agreed, we have edited the text to reflect the findings of the cited paper.

Page 5, l 38: I am not sure if SCT can be regarded as „widely investigated“ in MS. In fact it has been a major focus of one of the co-authors.

- We have edited that sentence to reflect that it is the most commonly investigated theory rather than widely investigated.

Page 5, l 44: Please indicate how long activity behavior change was maintained in the internet study

- 3 months, the paper has been amended accordingly.

Page 8, l 12: Was the 6MWT primary outcome at week 12, 24 and 26?

- Yes, this has been clarified in the paper.

Page 9, l 40: Was fidelity in relation to the SCT intervention or tot eh physiotherapy classes analysed?

- Both, and this has been clarified in the text.

Page 13, ll 18 and table 2: Please clarify how aerobic fitness scores was derived and indicate how strength was assessed.

- This has been added to the text.

Page 15 table 4: How do authors explain while in both groups intervention effects were less after 24 weeks compared to week 12 but increased again to week 36? It might be just due to the changing sample size. While a third of the patients was lost at week 36 i would refrain from presenting data in parallel with that differing numbers. Otherwise authors might select only the cohort with complete data.

- What we previously called the per-protocol analysis selected only the cohort with complete data on all time points. We have re-named that as a “secondary analysis” consistent with comments from reviewer 2. Informally participants reported that being measured at week 24 was a prompt to re-start exercising and this point has been added to the discussion while endeavouring to retain its brevity.

Page 16 l 43: Please indicate number of patients for per protocol analysis

- This has been added.

Page 18, l 40: Topic of the paper is not design of the next trial.

- The purpose of this pilot trial was to investigate feasibility and preliminary efficacy and to derive estimates of effect to inform a definitive RCT.

Page 19, l 27: Where are data on walking speed given?

- Data on TUG are presented in tables 3, 4 and 5.

Page 20, l 3: Based on which data do authors conclude that delivery of the SCT education was adequate.

- We video and audio recorded the sessions and an assessor independent to the trial read the intervention manuals and cross checked with the video to ascertain if the sessions were delivered as intended. We have edited the fidelity section of the method to address this comment and clarified in the results that all sessions were delivered as per the intervention manual.

Page 20, l 44: If authors do not report pedometer data it is difficult to judge these as a strength.

- The objective measures referred to here are the 6MWT and TUG – this has been clarified in the text.

Page 21, l 3: Conclusion omits the failure oft he primary endpoint.

- As this was a pilot study, we cannot conclude that there is a failure of the primary endpoint as the study was not powered to do this. We have added to the conclusion that there was no difference in effect using the intention to treat analysis.

Reviewer: 2

Reviewer Name: Johnny Collett

Institution and Country: Movement Science Group, OxINMAHR, Oxford Brookes University, UK

Please state any competing interests or state 'None declared': None declared

Please leave your comments for the authors below; Please find attached

This is a useful study that seems well conducted and investigates an important area of MS rehabilitation. The study essentially tested the feasibility of adding a social cognitive theory behaviour change intervention to an exercise intervention of aerobic and strength training, using a randomised design, in sedentary people with MS that may have mobility problems. The study found it was feasible to deliver and evaluate the intervention. However, from the result, currently presented, it is difficult to evaluate the estimates of potential superior efficacy of incorporating SCT.

- Thank you for these positive comments, we agree that the current pilot study results cannot be used to establish superior efficacy and propose to move to a definitive trial based on these data.

My main concerns are:

The analysis approaches used in the study are not clear (Below are example NICE definitions)

The study does not seem to strictly follow intention-to-treat analysis 'An assessment of the people taking part in a trial, based on the group they were initially (and randomly) allocated to. This is regardless of whether or not they dropped out, fully adhered to the treatment or switched to an alternative treatment...'

It appears the 27 people excluded after randomisation were not included and, according to figure 1 only 54 at week 12, 42 at week 24 and 43 at week 36 were included in analysis. Whilst, the 'intention-to-treat' approach used seems appropriate, clarification is needed in its description.

- Thank you for this helpful comment. We have edited the analysis section of the method to describe more fully the approach to intention to treat analysis.

More importantly I recommend removing the per- protocol analysis as feel it is not warranted in the context of this feasibility study. Per- protocol:

'A comparison of treatment groups in a trial that includes only those patients who completed the treatment they were originally allocated to. If done alone, this analysis leads to bias.'

The analysis reported is based on whether participants attended assessment or not, rather than completed the treatment as intended. I am not sure of the value of this analysis and would say it actual distracts from more valuable insights of the study.

- Thank you for this helpful comment. We do agree that we have incorrectly named the analysis as "per-protocol". We have re-named this analysis as a "secondary analysis of a cohort with data for three of four assessments" and refer to it later in the paper as "secondary analysis". As this is a pilot study, exploratory analysis is warranted (and is in fact suggested by reviewer 1) therefore we propose leaving this re-named analysis in the paper. This analysis is critical for understanding the efficacy of the interventions in those who completed the trial AND all assessments for a picture of the effect under ideal vs. real conditions of administration.

The reporting of the outcome in results is a bit confusing and sometimes redundant. Most of the results appear to be focused on within group change, which does not greatly add to the literature. The interesting outcome results are the between group analysis and whilst difference between groups may be small, this would be expected given that both groups are exercising and outcomes are primarily walking. However, these small effects could potentially be important in terms of long term participation in exercise.

- We fully agree that the between group analysis is the more useful analysis and have shifted the focus of the results towards those results.

However, given this is a pilot study, we feel that it is worth reporting all interesting results, as these could serve as useful hypotheses to test in the definitive trial.

- We respectfully disagree that this information is already in the literature, very few studies report any follow up, and our follow up at 24 and 36 weeks do provide new information as to whether improvements in walking can be maintained after an intervention.

I feel the 6minute walk was not the optimal to evaluate the main aim of the intervention (added SCT)

to enable people to meet MS exercise guidelines. In my opinion the most interesting and important outcome findings is that the intervention did indicate encouraging results for meeting guidelines at that at 10weeks (68% SCT v 50% control). Longer term follow up of this or other exercise physical activity measures) would have been interesting. Indeed the outcomes used in this study are mainly focused on impairment/performance. I believe discussion of this and the benefits of a longer term follow up in relation to future studies) is important (we pretty much know exercise will benefits walking, the challenge is longer term participation with the goal of sustained benefit to symptoms, health and QoL).

- Thank you for this comment. We agree that the difference in proportions meeting the guidelines is interesting and that sustaining outcomes is important. This is consistent with our reporting of within group differences at 24 and 36 weeks which provides interesting new insights into how we can sustain exercise behaviour and hence improvements in walking. We do propose to continue to use the exercise logs up to six month follow up in the definitive trial and have amended the discussion to reflect this. Because the protocol defined 6MWT as the primary outcome we propose leaving it as the primary in this paper; this would be disingenuous if we removed this as our primary endpoint. Table 2 should report between group differences (effect sizes and (95% CI)), not within.

- Table two presents information on the concepts underpinning the intervention, i.e. the intervention aimed to improve PA, strength and fitness and these data are in effect a “manipulation check” as to whether the intervention changed what it was intended to do. These data provide information as to whether the exercise intervention was successful in improving strength and aerobic fitness and is part of the feasibility/fidelity analysis of the paper

I recommend mobility outcomes be reported in one table, basically table 3 with the addition of the between group treatment at effects of table 5 (reported as between group effect size with 95% CI)

- Thank you for this suggestion, we have tinkered with various layouts and formats for a combined table – we feel that combining these data into one table makes it very difficult to read and respectfully suggest leaving the tables separate.

In the trial registration primary outcome was all walking mobility measures whereas the study reports 6min walk as the primary. This should be made consistent.

- Apologies, the heading “secondary outcomes” has been removed to ensure consistency between protocol and this paper

Below are some further comments and suggestions

In the UK NIHR have distinct definition of feasibility and pilot studies (The title of the paper referring to pilot and objective being feasibility might be problematic for some people here)

- At the request of the editor we have named this as pilot and used the CONSORT extension for pilot and feasibility studies

Please provide numbers analysed in abstract as per CONSORT

- This has been added

In methods more detail could be added to study design section (page 6) (eg 2 arm, parallel (1:1))

- This has been added

Whilst, the sample size seems appropriate to fulfil the feasibility aims of the study, I am not quite sure how the authors derived their estimate and would like some clarification. It appears to be based on a between group mean difference in 6min walk of 36m +/- 48.2.

I am not sure how this estimate was derived from the paper cited (eg the between group treatment effect), but also as a feasibility study I would expect a function of the study would be to provide estimates of effect rather than detect a difference.

- As you correctly point out, the function of this pilot study is to provide an estimate of effect.

Consistent with the CONSORT extension for pilot trials we provide the rationale for the numbers in the pilot trial. We used data from a large cohort study that reported the degree of change that is meaningful to patients and used that to guide numbers in the pilot.

In addition a sample size of 72 people is stated in the protocol paper (to allow for drop outs). This should be made consistent, did the study not fully recruit (consideration for feasibility) or were dropouts less than expected (leading to protocol change).

- We have edited the paper to provide clarification on this point, on recruiting 72 people it became apparent that recruitment per region did not lead to sufficient numbers to run a class. We continued to recruit to 92 and at that point made a pragmatic decision to cease recruitment given the attrition of 27 people.

JN generated the random allocation sequence yet JN was also blind to allocation, this is contradictory at face value and needs clarification?

- Apologies for this error, JN was not blinded to allocation, CS who performed the analysis was. This has been amended

Details of the track/area used for 6min walk (eg length of circuit, were people allowed to rest) would Add

- This has been added

The authors report hip pain as the only adverse, was this this the only related AE, as the flow diagram indicates several relapse across groups (which could be considered AEs even though not related)

- There was 1 AE (hip pain) that was likely associated with the intervention, and other AEs (relapses) that we unlikely associated with the treatment.

Even with the Protocol paper I would find it very difficult to replicate the intervention, perhaps a description according to TIDieR could be added as a supplement. Making available the Manual of operating procedures or the intervention Manual would be very useful to anyone wishing to utilise the intervention.

- Thank you for this suggestion. As much of the information is included in the protocol paper (open access) we have provided the TIDieR template as an appendix that brings together the information in this paper and the protocol in one location

I feel the discussion would benefit from the addition of how this intervention may fit within current health or 3rd sector services, in particular in Ireland and perhaps wider.

- Thanks for this suggestion, we feel it has particular application for primary care and this has been added

VERSION 2 – REVIEW

REVIEWER	Johnny Collett MORes, OxINMAHR, Oxford Brookes UNiversity, UK
REVIEW RETURNED	25-May-2017

GENERAL COMMENTS	The review has essentially addressed my concerns and whilst the authors and I disagree on a couple of points this does not distract from the over all merit of the work (although, I would still like to see between group effect sizes (eg Cohen's d with CI:95%). This is a relevant and useful study.
--

VERSION 2 – AUTHOR RESPONSE

We agree with the reviewer that representing the likely magnitude of the treatment effect is important and have therefore included an estimate of Hedges G for all three walking measures and added information on this in the methods, results and discussion sections. The use of Hedges G is consistent with the analysis presented in the published paper for the secondary measures and will allow cross comparison between the two papers. We hope that you now find the paper ready for publication.